# The Mediating Role of Psychological Balance on the Effects of Dietary Behavior on Cognitive Impairment in Chinese Elderly

**DOI:** 10.3390/nu16060908

**Published:** 2024-03-21

**Authors:** Yating Chen, Lingling Zhang, Xiaotong Wen, Xiaojun Liu

**Affiliations:** 1Department of Social Medicine and Health Management, School of Health Management, Fujian Medical University, Fuzhou 350122, China; yating_tina@fjmu.edu.cn; 2Department of Epidemiology and Health Statistics, School of Public Health, Fujian Medical University, Fuzhou 350122, China; 2210220176@fjmu.edu.cn; 3Department of Social Medicine and Health Management, School of Public Health, Wuhan University, Wuhan 43000, China; 2020103050014@whu.edu.cn

**Keywords:** dietary diversity, dietary patterns, cognitive impairment, psychological balance

## Abstract

Background: Cognitive impairment, a significant problem in older adults, may be associated with diet. This study aims to examine the association between the dietary diversity score (DDS), dietary pattern (DP), and cognitive impairment in elderly Chinese. This research further explored the role of psychological balance (PB) as a mediator in the relationship between diet and cognitive impairment. Methods: A total of 14,318 older adults from the Chinese Longitudinal Healthy Longevity Study (CLHLS) in 2018 were included. Latent class analysis (LCA) was used to identify patterns in seven food varieties. Binary logistic regression models were used to determine factors associated with the DDS, DP, and cognitive impairment. The multiple mediation effect model was evaluated using model 6 in the PROCESS version 3.5 program. Results: Among the participants, 4294 (29.99%) developed cognitive impairment. Compared to people in food variety group two or lower, people with a high dietary diversity score (DDS) had lower odds of cognitive impairment. Compared to DP1, DP2 (OR = 1.24, 95%CI = 1.09 to 1.40) was associated with a higher risk of cognitive impairment, and DP4 (OR = 0.79, 95%CI = 0.69 to 0.89) was associated with a lower risk of cognitive impairment. PB mediated the relationship between DDS, DP, and cognitive impairment, with a mediating effect of 27.24% and 41.00%. Conclusions: A DP that is rich in fruits, vegetables, red meat, fish, eggs, beans, nuts, and milk was related to a lower risk of cognitive impairment. PB has an indirect impact on cognitive impairment. Our findings underscore the importance of promoting a diverse diet, which may contribute to a lower risk of cognitive impairment in older adults. The PB of the elderly should also be taken into consideration.

## 1. Introduction

Cognitive impairment is defined as a decreased level of cognition abilities in older adults, which affects the ability of the patients to remember, learn, concentrate, and make daily decisions. Worldwide cognitive impairment prevalence rates range from 3% to 53.8%, depending on the criteria deployed to measure the condition [1]. Globally, cognitive impairment is common in older populations. Individuals with cognitive impairment have a higher risk of developing dementia (most frequently Alzheimer’s disease [2]) and have higher death rates [3]. Studies have shown that a diagnosis of cognitive impairment in healthy older adults represents a transitional period before the onset of dementia. Around 15% to 35% of older adults will progress to dementia each year [4], and 46% of mild cognitive impairment patients develop dementia within three years of diagnosis [5]. However, cognitive impairment is thought to be a reversible condition [6]. Deterioration can be better avoided if high-risk older persons with cognitive impairment are identified early. Therefore, identifying modifiable factors that slow the progression of cognitive impairment in older adults is critical to preventing or delaying dementia.

Malnutrition and psychological imbalance are two health issues affecting older people that have been gaining more attention in recent years. These two issues are frequently linked to cognitive impairment and are thought to be detrimental [7]. Clinical trials, observational research, and laboratory research all provide evidence of the modifying role that nutrition plays in cognitive performance throughout life [8] and have been incorporated into strategies for preventing or delaying the onset of cognitive impairment [9]. According to research on dietary patterns, memory is improved by a low-carbohydrate diet [10] whereas the Mediterranean diet decreases the likelihood of cognitive impairment progressing to dementia [11]. Two assessments of dietary patterns and cognition in non-MCI populations found that the Mediterranean, Dietary Approaches to Stop Hypertension (DASH), and Mediterranean-DASH Intervention for Neurodegenerative Delay (MIND) diets, which are all plant-based diets with moderate to high intakes of fish, are all associated with improved cognitive outcomes [12,13]. Regarding specific diets, epidemiological data bolster the belief in Western cultures that a comparatively higher intake of fruits and vegetables, which are high in polyphenols, reduces the incidence of neurocognitive decline and dementia [14,15]. Many studies have confirmed the positive effects of fish on memory [16,17]. However, additional research has not provided enough evidence to substantiate these findings. Thus, more research is required while the role of confounders and underlying mechanisms are not fully understood.

When analyzing the association between diet and cognition, it is crucial to consider the impact of additional protective factors such as psychological balance (PB), which may affect the relationship. Previous studies have also demonstrated that psychological distress has adverse effects on dietary intake [18]. Anxiety and depression symptoms linked to psychological imbalance can be caused by dietary imbalance and low dietary diversity [19]. Specifically, the frequent consumption of meat or fish has been linked to PB [20]. Psychological balance symptoms including anxiety and depression have been reported to be substantially linked to cognitive impairment in several studies [21] and to be significant predictors of cognitive impairment [22]. Wetherell et al. [23] examined the relationship between anxiety and cognition in a population of 704 healthy elderly people. In the preliminary assessment, anxiety was linked to worsening memory; nevertheless, anxiety was not found to be a predictor of cognitive decline in that study. According to studies by Freire et al. [24], there is no evidence of a significant correlation between depression, anxiety disorders, and a deterioration in cognitive performance in those over the age of 55.

The evolution of pharmacological therapies is an extremely difficult endeavor that comes with a high cost and demands substantial resources. In contrast, preventive measures can be implemented more broadly and are less expensive. Dietary scores quite highly among such strategies, and has the added benefit of being accepted by most of the population as more “benign” and potentially free of side effects. Given that dietary intervention is an enticing strategy in the fight against cognitive impairment [25], its potential utility for reducing cognitive impairment is vital. Therefore, this study investigated the intricate relationship between diet and cognitive impairment using the Chinese Longitudinal Healthy Longevity Survey (CLHLS).

## 2. Materials and Methods

### 2.1. Study Design and Samples

The study used cross-sectional data from the Chinese Longitudinal Healthy Longevity Survey (CLHLS), which was co-developed by the National Institutes of Development of Peking University’s Research Center for Healthy Aging and Development and the Chinese Center for Disease Control and Prevention in 2018. The CLHLS is a nationally representative study that explores aspects related to human health and lifespan among older persons in most Chinese provinces. The study used a structured questionnaire conducted by trained investigators in the participant’s home. Spouses or other close family members were interviewed as surrogate respondents in cases where the respondents were unable to answer questions. The Peking University Research Ethics Committee approved the CLHLS study (IRB00001052-13074), and all participants signed written informed consent [26]. A total of 15,875 participated in CLHLS in 2018. However, 103 participants were excluded as their age was less than 65 years and 1453 participants were eliminated due to missing data on psychological balance information.

### 2.2. Measures

Sociodemographic information. Basic characteristics of the respondents were taken into account in the analysis to control for variance in the use of informal care based on gender (male vs. female), age (65–74 years, 75–84 years, 85–94 years, and ≥95 years), educational level (below primary school, primary school and junior high school, or above), marital status (married vs. others including “ divorce”, “widowed”, and “never married”), current residence (county vs. village), living alone (yes vs. no), occupation before age 60 (agriculture vs. non-agricultural), self-assessed economic situation (wealthy, average and poor), self-assessed health (good vs. poor), and social participation (yes vs. no).

Dietary diversity. Referring to previous studies [27,28], we employed the food frequency questionnaire in the CLHLS to calculate a dietary diversity score (DDS). The state of the dietary groups’ intake covers vegetables, fruit, red meat, fish, eggs, beans–nuts, and milk. DDS has been widely used to assess dietary diversity with good reliability and validity [29,30]. There were options for intake (yes or no) to ask about food frequency. The choice for vegetables and fruits was four optional responses “daily/almost daily”, “frequently”, “sometimes”, and “rarely/never”. Vegetables and fruits were defined as sufficient if answered daily/almost daily and frequently were coded as 1. The reverse was valued as 0. The choice for red meat, eggs, milk, and beans–nuts was five optional responses: “almost daily”, “weekly”, “monthly”, “sometimes”, and “rarely/never”. Red meat, eggs, milk, and beans–nuts were classified as sufficient if answered almost daily/weekly and were coded as 1. Conversely, they were coded as 0. The DDS calculates the score of the adequate intake of seven food groups; values vary from 0 to 7. The highest score is 7, meaning that all seven food groups have been sufficiently taken, while the lowest score is 0, indicating that none of the seven groups have been adequately consumed.

Psychological balance. Using the selection and calculation rules for indicators from previous research [31,32,33], psychological balance (PB) was established. Specifically, this study assessed PB by selecting a series of questions from the questionnaire “Assessment of the current situation and emotional characterization of the personality”. One portion of the questionnaire captures older adults’ subjective opinions on their general quality of life and standard of living across time. These opinions are relatively stable, and they can be a measure of psychological balance. Positive and negative emotions are reflected in three questions each. Positive emotions: “How do you feel about your life now?”, “Are you able to think about whatever is happening to you?”, and “Do you feel energized?”. Negative emotions: “Do you feel ashamed, regretful, or guilty about things you’ve done?”, “Do you get angry at people or things you don’t like around you?”, and “Do you often feel that no one around you is trustworthy?”. Responses to questions reflecting positive emotions were given inverted scores such as “very good”, “good”, “fair”, “bad”, and “very bad” were given a score of 5–1 in turn; positive scores were given to responses to questions reflecting negative emotions such as “always”, “often”, “sometimes”, “rarely”, and “never” were given a score of 1–5 in that order. This was undertaken to standardize the direction of data measurement and to make calculation easier. Therefore, the range of both positive and negative emotions was 3 to 15 points. The total score ranged from 6 to 30 points, with higher scores indicating better PB. The PB in our study ranged from 7 to 30, with a mean and standard deviation of 22.70 ± 3.56.

Cognitive impairment. The cognitive function of the seniors was measured by the Mini-Mental State Examination (MMSE), which was created by Folstein [34] and has been adapted for the cultural and socioeconomic conditions in China [35]. The scale includes 24 items spread across six dimensions, with a total score of 0 to 30 points: five items for Orientation, three for Registration, one for Naming, five for Attention and Calculation, three for Recall, and seven for Language. The question “Name the number of food items in 1 min” had a maximum score of 7 points. The other 23 items received a maximum of 1 point each. Better cognitive function was predicted by higher scores, with cognitive impairment recognized at a score below 24.

### 2.3. Statistical Analysis

Dietary patterns were performed in Mplus version 8.3 (Muthen & Muthen, Los Angeles, CA, USA), and correlation and mediation analyses were conducted on SPSS version 25.0 (IBM Corp, Armonk, NY, USA) at a significance level of 0.05. Four phases made up the data analysis. First, frequencies and proportions were employed to conduct a descriptive study of the demographic characteristics, seven chosen food varieties, and DDS. The second step was to carry out a latent class analysis (LCA) in seven food varieties to identify the dietary pattern (DP) of elderly Chinese. Third, the Chi-square test was conducted to compare the cognitive impairment across the specific diet, DDS, and DPs. Then, binary logistic regression was used to analyze the correlation between the specific diets, DDS, DPs, and cognitive impairment; odds ratios (ORs) with associated 95% confidence interval (CI) and *p*-values were presented in the model. Finally, a chain mediation test was performed by model 6 in PROCESS 3.5, and the significance of the mediation effect was tested by sampling 5000 times by bias-corrected percentile bootstrap.

## 3. Results

### 3.1. General Demographic Characteristics of the Study Participants

Table 1 shows the general demographic characteristics of Chinese older adults according to cognitive impairment. A total of 14,318 participants were included in the study, the overall rate of cognitive impairment was 29.99%, among these, 1304 female participants had cognitive impairment (37.93%), 2158 participants aged 95 and over had cognitive impairment (66.26%), 3172 participants with a primary school education or lower had cognitive impairment (46.21%), 3426 of the unmarried, divorced, and widowed participants had cognitive impairment (42.48%), 557 of the participants with poor self-assessed economic situation had cognitive impairment (37.89%), and 4020 of those without social participation had cognitive impairment (34.11%). The mean age of the sample was 79.13.

### 3.2. The Seven Food Varieties and Dietary Diversity

Figure 1 shows the characteristics of the percentages for the seven food varieties. Among all of the participants, 89.41% of them consumed vegetables daily/almost daily and frequently, 76.86% of them consumed red meat almost daily/weekly, 72.08% of them consumed eggs almost daily/weekly, and 51.44% of them consumed beans–nuts almost daily/weekly. In contrast, those who had an almost daily/weekly milk intake presented low percentages at 39.04%. Figure 2 illustrates the constitution ratio of the DDS. Among those respondents, 19.13% had two or lower food varieties, 15.59% had three food varieties, 19.07% had four food varieties, 19.77% had five food varieties, 15.64% had six food varieties, and 10.80% reported seven food varieties.

### 3.3. Latent Class Analysis of Dietary Patterns

As Table 2 shows, latent class models with 1–6 classes were calculated. The 4-class solution (AIC: 112,421.239; BIC: 112,655.886; ssaBIC: 112,557.371; entropy: 0.695) was chosen as the final model for this investigation due to its lower AIC, BIC, and ssaBIC, higher entropy, and LMR-LRT and BLRT <0.001. Ultimately, four DPs were identified. The estimated class-specific response probabilities, which display the four DPs in the 4-class solution, are shown in Figure 3. In addition, the response probabilities of seven food groups in four dietary patterns is shown in Appendix A.

DP1 was characterized by individuals with a high probability of consuming vegetables and red meat, with low possibilities for sufficient eggs, beans–nuts and milk including 23.62% of the sample. DP2 was defined as those individuals (20.39% of the sample) who had a low probability of consuming enough fish and beans/nuts and a high probability of ingesting vegetables and eggs. DP3 accounted for one-third of the sample (32.57%), among these were individuals with little probability of attaining appropriate fruits and high possibilities of taking vegetables, red meat, fish, eggs, and beans and nuts. DP4 was defined as the individuals who, comprising 23.42% of the sample, had the highest rate of consuming a wide variety of food.

### 3.4. Chi-Square Test of Dietary and Cognitive Impairment

In Table 3, it was found that vegetables (χ^2^ = 367.374), red meat (χ^2^ = 142.824), eggs (χ^2^ = 122.027), beans–nuts (χ^2^ = 171.180), fish (χ^2^ = 12.987), fruits (χ^2^ = 45.962), and milk (χ^2^
*=* 11.436) were the influencing factors of cognitive impairment (*p* < 0.05). Chi-squared tests illustrated that DDS (χ^2^ = 260.706/256.236, *p* < 0.001) and DPs (χ^2^ = 219.698/157.422, *p* < 0.001) were significantly associated with cognitive impairment (Table 4).

### 3.5. Binary Logistic Regression Analysis Testing the Association between the Dietary and Cognitive Impairment

Table 5 shows the association between a specific diet and cognitive impairment. After adjusting for confounding factors, participants who consumed adequate vegetables (OR = 0.79, 95%CI = 0.72 to 0.87), fruits (OR = 0.51, 95%CI = 0.45 to 0.58), red meat (OR = 0.70, 95%CI = 0.64 to 0.78), and fish (OR = 0.74, 95%CI = 0.68 to 0.81) were less likely to have cognitive impairment compared to those with insufficient intake. Table 6 presents the association between DDS, DPs, and cognitive impairment. Compared to people in two or lower food varieties, people with a high DDS had lower odds of cognitive impairment (3: OR = 0.79, 95%CI = 0.69 to 0.91; 4: OR = 0.75, 95%CI = 0.65 to 0.86; 5: OR = 0.66, 95%CI = 0.57 to 0.75; 6: OR = 0.59, 95%CI = 0.51 to 0.69; 7: OR = 0.54, 95%CI = 0.45 to 0.65). Compared to DP1, DP2 (OR = 1.24, 95%CI = 1.09 to 1.40) was associated with a higher risk of cognitive impairment and DP4 (OR = 0.79, 95%CI = 0.69 to 0.89) was associated with a lower risk of cognitive impairment.

### 3.6. Pearson Correlation Analysis and Chain Mediated Effects Test

The results of the Pearson correlation analysis revealed that the DDS and DP were positively correlated (*r* = 0.862, *p* < 0.01), while PB and DDS as well as PB and DP were positively correlated (*r* = 0.177, *p* < 0.01; *r* = 0.150, *p* < 0.01, respectively). Cognitive impairment and DDS, cognitive impairment and DP, along with cognitive impairment and PB were negatively correlated (*r* = −0.142, *p* < 0.01; *r* = −0.105, *p* < 0.01; *r* = −0.233, *p* < 0.01, respectively). Table 7 provides the detailed results.

Based on the correlation of DDS, DP, PB, and cognitive impairment, we used multiple mediation analysis to examine the mediating role of DDS, DP, and cognitive impairment in the Chinese elderly. With DDS and DP as the independent variable (X), PB as the mediating variable (M), and cognitive impairment as the dependent variable (Y), a chain mediation model was constructed. The mediation path model is shown in Figure 4 and Figure 5. The path coefficient indicates whether each relationship in the model is significantly positive or negative. DDS indirectly affects cognitive impairment through mediation: PB in Figure 4. As shown in Figure 5, DP indirectly affects cognitive impairment through mediation: PB.

To test whether the mediation effects of PB were significant, we designed an autonomous estimation procedure for the samples. The effect is the most when the pathway coefficient of the 95%CI does not include 0. Table 8 and Table 9 list the total, direct, and indirect effects. As shown in Table 8, when mediating factors were incorporated into the model, the direct effect of DDS on cognitive impairment was significant (B = 0.029, 95%CI = 0.163 to 0.277), accounting for 73.79% of the total effect, while PB was found to indirectly affect cognitive impairment through the significant mediating pathways: PB (B = 0.008, 95%CI = 0.066 to 0.099), accounting for 27.24% of the total effect. As shown in Table 9, when mediating factors were incorporated into the model, the direct effect of DP on cognitive impairment was significant (B = 0.043, 95%CI = 0.050 to 0.218), accounting for 59.04% of the total effect, while BF was found to indirectly affect cognitive impairment through the significant mediating pathways: PB (B = 0.012, 95%CI = 0.070 to 0.116), accounting for 41.00% of the total effect.

## 4. Discussion

Diet is a modifiable risk factor with major public health implications [36]. Maintaining a varied diet and choosing a quality dietary patterns (DPs) are vital for obtaining enough nutrients [37,38]. Furthermore, the association between dietary diversity score (DDS) and DPs and the subsequent risk of cognitive impairment is uncertain. Therefore, in this study, we examined the association between diet and cognitive impairment in older adults, introducing psychological balance (PB) as a mediator. PB contributed significantly to our understanding of the key strategies for enhancing this population’s diet and halting the onset and progression of cognitive impairment in the long term. In this study, we adopted seven food groups and calculated the intake of the total number of food varieties to assess the DDS. We obtained four DPs by applying the LCA to identify the DPs in a representative sample. According to the findings, certain diets (vegetables, fruits, red meat, fish, eggs, beans–nuts, and milk) are linked to cognitive impairment. Similarly, DDS and DPs are related factors that cause cognitive impairment.

As anticipated, the results from our study provide evidence, before and after controlling for potential confounding variables, that older people with an adequate intake of vegetables, fruits, red meat, fish, and milk had considerably lower detection rates of cognitive impairment. In a large prospective cohort research with a six-year follow-up, people 65 years of age and older who consumed a large amount of vegetables, but not fruits, had a reduced rate of cognitive decline [39]. We reviewed a recent systematic study that found eating more vegetables was linked to a decreased risk of dementia and a slower rate of cognitive decline as people aged [40]. A previous systemic review found that increased vegetable and fruit intake was related to a reduced risk of cognitive impairment [41], which is consistent with the findings of our study. In contrast, another systematic review concluded that there was insufficient and inconsistent evidence to relate the consumption of fruits and vegetables to dementia and mild cognitive impairment [42].

Evidence from early studies has shown inconsistent associations between the consumption of meat and cognitive impairment. In the 85+ Cohort Study of Newcastle (UK), low cognitive function was linked to a high red meat intake [43]. According to data from 194 cognitively healthy participants in the Uppsala Seniors cohort research, eating less meat and meat products was associated with higher cognitive evaluations [44]. In a large prospective cohort study, Jiang et al. [45] discovered a positive association between the consumption of red meat, fresh and preserved, and the risk of cognitive impairment. However, other studies have shown different results. According to the Maine-Syracuse Longitudinal Study, an intake of more meat was prospectively linked to improved cognitive scores [46]. Similarly, our findings demonstrate that consuming enough red meat lowers the possibility of developing cognitive impairment. The fact that red meat contains heme, which makes up 95% of human functional iron, may explain the link [47], and may indicate that a diet low in red meat may raise the risk of cognitive impairment. In addition, the CLHLS, which investigated participants over 65 years and older, discovered no connection between consuming more meat and a greater risk of dementia [48].

In our study, we discovered that a sufficient fish intake was associated with a lower risk of cognitive impairment. There is also evidence of the protective effect of fish consumption on cognitive decline [49]. A meta-analysis combining data from one French and four U.S. cohorts indicated that consuming higher quantities of fish was associated with a slower deterioration of cognitive functioning [16]. Another meta-analysis also revealed a negative correlation between fish intake and the risk of dementia and Alzheimer’s disease [50]. Additionally, cohort research covering 1566 Chinese individuals over a mean of 5.3 years indicated that overall fish consumption was notably associated with a slower rate of cognitive deterioration in persons 65 years and above [51]. Dairy products and milk may reduce the risk of cognitive decline, either directly or via their mediating effects on cardiometabolic health. In a cross-sectional investigation, Muñoz-Garach et al. found that the consumption of whole-fat milk was associated with a lower risk of cognitive impairment [52]. Our results indicate that the consumption of enough milk is connected with a lower incidence of cognitive impairment, which is consistent with the results of previous studies [53]. As a potential explanation, Camfield et al. [54] proposed that specific bioactive peptides could be advantageous in maintaining normal brain function as people age. To follow-up on this theory, data from seven cohort studies and one randomized controlled trial were collated in a 2017 systematic review using observational research methods. The review concluded that the evidence available was insufficient to establish a causal link between the intake of milk by older adults and cognitive disorders [55].

Diet is a multidimensional exposure, so it is difficult to determine whether a single food is accountable for the prevalence rate of a specific disease or symptomatology [56]. Thus, we measured the total healthy eating behaviors in our research. Long-term research by Zheng et al. [57] indicated that individuals with higher baseline DDSs had a lower chance of cognitive impairment than those with lower DDSs. Additionally, during long-term follow-up, they discovered that a higher DDS may slow the rate of cognitive deterioration. Song et al. [58] provided more context for these findings by demonstrating that regardless of the baseline DDS, maintaining a high DDS or enhancing the DDS could reduce the risk of cognitive impairment. Consistent with previous studies, our data indicate that DDS ≤2 subjects were more likely to experience cognitive impairment. One probable reason could be that the brain is highly vulnerable to oxidative damage and older individuals frequently have a low antioxidant status [59]. Due to its connection to elevated levels of oxidative stress, a low-diversity diet may subsequently raise the risk of cognitive impairment in the elderly [60].

Considering the intricate interactions between different nutrients and food, DPs have been connected to cognitive impairment in multiple studies. According to a study by Shikh et al. [61], a whole-food diet was not associated with a decreased risk of cognitive impairment. In contrast to this study, we discovered that DP4—which is high in fruits, vegetables, red meat, fish, eggs, beans, nuts, and milk—was associated with cognitive impairment when compared to DP1. The findings of some epidemiological studies are consistent with our findings, despite the differences in the dietary pattern analysis. Based on a reduced rank regression analysis of 2311 Chinese elderly people aged 60–88 years, a high consumption of vegetables, fruit, legumes, and nuts is linked to improved memory and language function and a lower risk of cognitive impairment [62]. The positive effects of antioxidants and vitamins C, D, and E, which are abundant in fruits, may be the reason for the association between DP4 and cognitive function. Increased consumption of antioxidants may avert age-related cognitive decline since brain tissue, which is especially susceptible to free-radical damage, has low amounts of endogenous antioxidants [63]. Nutrients like vitamin B12 [64], choline [65], and protein [66] are found in eggs, milk, and dairy products and are linked to improved cognitive function. A 2019 systematic review of six studies indicated that dairy products could guard against cognitive aging [67]. Nevertheless, a study conducted in the same year with 3835 Americans aged 65 years old and older did not find a similar link between egg consumption and cognitive health [68]. Based on our research, DP2 subjects with a higher intake of legumes, milk, and eggs were more likely to be cognitively impaired than DP1 subjects. Elderly people may also have a decreased capacity for nutrient digestion and absorption. Furthermore, DP2 subjects consume less fish and red meat than DP1 subjects, which may deprive the body of vital nutrients and increase the risk of cognitive impairment.

In previous studies, there was a correlation between the DDS, DP, and cognitive impairment. This aligns with our presumptions. Given the importance of social life in current Chinese society, people are becoming more and more engaged in a healthy lifestyle including PB. However, previous studies assessed the DDS, DP, and cognitive impairment as multiple and separate variables and only a few studies focused on influencing mechanisms like PB. PB may be related to protective factors concerning diet and cognition. There is a relationship between diet and PB [18]. Studies have indicated that that PB is a reliable indicator of cognitive impairment [22]. By comprehending the crucial role of PB, it becomes possible to implement nutritional interventions from a psychological perspective (unlike the perspective of previous studies, which mainly explored the factors linked to obesity and metabolism as moderators of diet to cognition [69]) and assist with managing cognitive impairment. Our study found that PB mediated the relationship between DDS, DP, and cognitive impairment with a mediating effect of 27.24% and 41.00%, respectively. Nutritional imbalances and a lack of dietary diversification may contribute to psychological disorders in older adults. Maintaining a healthy microbiota may also be crucial for mental health, according to recent research on the gut–brain axis [70]. Therefore, a richer dietary diversity may contribute to better PB through healthier gut microbiota. When the psyche is in balance, it is easier for senior citizens to deal with their self-management of cognitive decline (e.g., paying attention to indicators of cognitive decline).

One of the strengths of this study was that the data covering 23 provinces or municipalities came from the CLHLS. These areas are inhabited by typical Chinese citizens but differ in terms of their geographic location, degree of economic growth, availability of public resources, and health indices. Furthermore, in this study, we used the LCA method to identify DPs, which provides an innovative perspective. However, our study has several limitations, which should be noted. Limited by the use of diagnostic tools, our study did not classify cognitive impairment in detail (e.g., mild cognitive impairment or different types of dementia). As this was a cross-sectional study, reverse causality and bidirectional associations cannot be ruled out. In addition, food frequency questionnaires that rely on the self-reported frequency of dietary consumption are vulnerable to recall bias. Bias could also have been generated by using surrogate respondents in our study. Moreover, we failed to account for the periods of time where the participants adopted specific dietary patterns. Finally, even though we took covariant changes and a wide range of potential confounding factors into account, residual and unmeasured confounding effects cannot be completely ruled out in this observational analysis. Thus, more components for a thorough evaluation must be included in future studies on diet and cognitive impairment.

## 5. Conclusions

In this study, we observed that maintaining a high dietary diversity could reduce the risk of cognitive impairment among the elderly Chinese population. We found that a DP that is rich in fruits, vegetables, red meat, fish, eggs, beans, nuts, and milk was related to a lower risk of cognitive impairment. We demonstrated the relationships between dietary diversity scores, dietary patterns, and cognitive impairment and provide empirical evidence that psychological balance has an indirect impact on cognitive impairment. Our findings underscore the importance of promoting a diverse diet, which may contribute to a lower risk of cognitive impairment in older adults. The PB of older adults should also be taken into consideration. Our findings confirm the requirement made by the Chinese Dietary Guidelines for the Elderly to consume a wide variety of foods including moderate amounts of fruits and vegetables and plenty of fish, poultry, meat, eggs, and beans. This study offers good evidence that encouraging healthy aging can be achieved with a scientific diet. Furthermore, future random controlled trials may be needed to evaluate whether dietary interventions that adopt dietary diversity scores and dietary patterns can effectively bypass or delay cognitive impairment.

## Figures and Tables

**Figure 1 nutrients-16-00908-f001:**
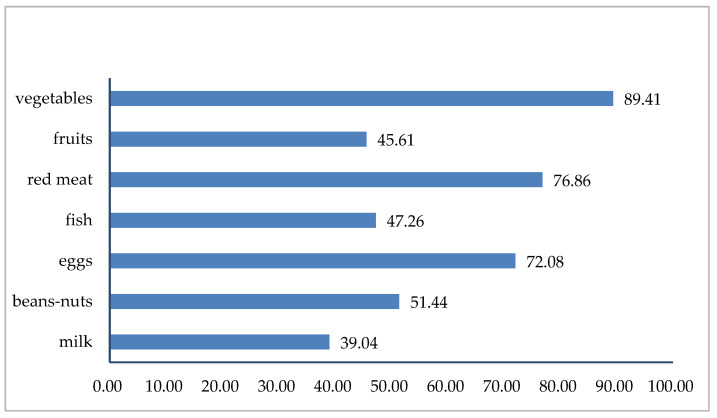
Percentages of the seven food varieties (%).

**Figure 2 nutrients-16-00908-f002:**
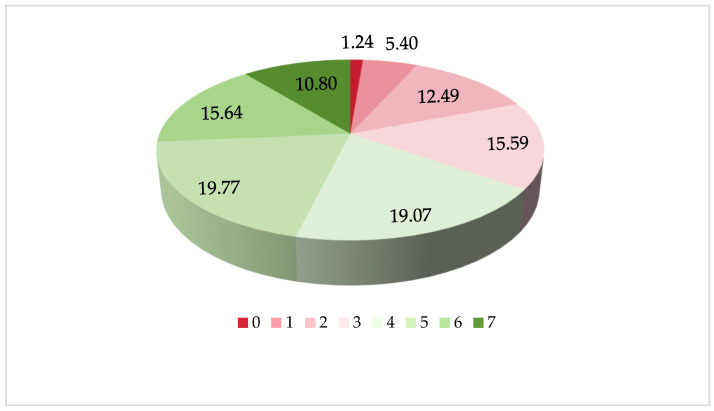
The constitution ratio for the dietary diversity (%).

**Figure 3 nutrients-16-00908-f003:**
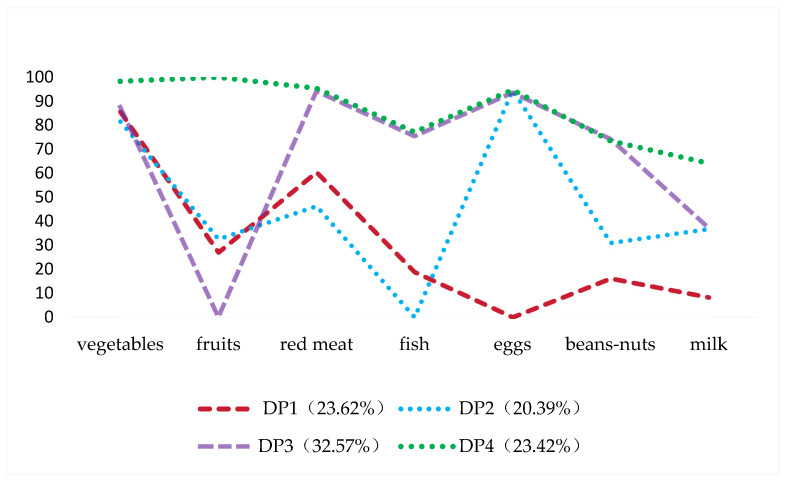
Estimated class-specific response probabilities for the seven food varieties.

**Figure 4 nutrients-16-00908-f004:**
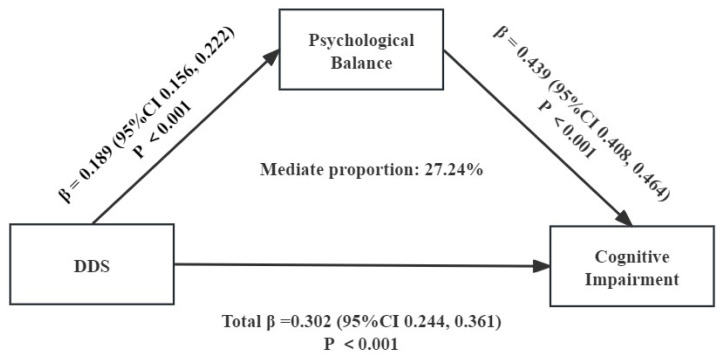
A chain mediation model through the association between dietary diversity scores, psychological balance, and cognitive impairment.

**Figure 5 nutrients-16-00908-f005:**
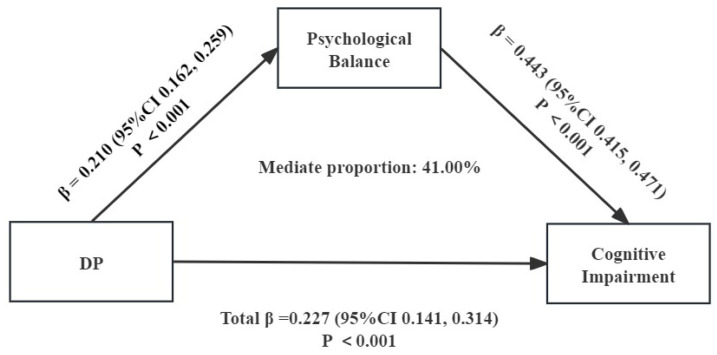
A chain mediation model through the association between dietary patterns, psychological balance, and cognitive impairment.

**Table 1 nutrients-16-00908-t001:** Basic information of the participants in the survey (n = 14,318).

Variables		Participants	Constitutional Ratio (%)	Cognitive Impairment
Gender	Male	6435	44.94	1304 (20.26)
	Female	7883	55.06	2990 (37.93)
Age (years)	65–74	3249	22.69	168 (5.17)
	75–84	4166	29.10	648 (15.55)
	85–94	3646	25.46	1320 (36.20)
	≥95	3257	22.75	2158 (66.26)
Educational level	Below primary school	6864	47.94	3172 (46.21)
	Primary school	4695	32.79	829 (17.66)
	Junior high school or above	2759	19.27	293 (10.62)
Marital status	Married	6253	43.67	868 (13.88)
	Others	8065	56.33	3426 (42.48)
Current residence	County	7955	55.56	2255 (28.35)
	Village	6363	44.44	2039 (32.04)
Living alone	Yes	2339	16.34	664 (28.39)
	No	11,979	83.66	3630 (30.30)
Occupation before age 60	agriculture	9124	63.72	3032 (33.23)
	non-agricultural	5194	36.28	1262 (24.30)
Self-assessed economic situation	Wealthy	2847	19.88	646 (22.69)
	Average	10,001	69.85	3091 (30.91)
	Poor	1470	10.27	557 (37.89)
Self-assessed health	Good	6737	47.05	1727 (25.63)
	Poor	7581	52.95	2567 (33.86)
Social participation	Yes	2533	17.69	274 (10.82)
	No	11,785	82.31	4020 (34.11)

**Table 2 nutrients-16-00908-t002:** Latent class analysis model fit statistics.

Model	AIC	BIC	ssaBIC	LRT *p* Value	BLRT *p* Value	Entropy
1-class	120,673.950	120,726.935	120,704.689	-	-	-
2-class	113,663.213	113,776.752	113,729.083	<0.001	<0.001	0.600
3-class	113,080.619	113,254.712	113,181.620	<0.001	<0.001	0.745
4-class	112,421.239	112,655.886	112,557.371	<0.001	<0.001	0.695
5-class	112,113.300	112,408.502	112,284.563	<0.001	<0.001	0.654
6-class	111,906.943	112,262.699	112,113.337	<0.001	<0.001	0.746

Abbreviation: AIC = Akaike information criterion, BIC = Bayesian information criterion, ssaBIC = sample size adjusted Bayesian information criterion, LRT = Lo–Mendell–Rubin likelihood ratio test, BLRT = bootstrapped likelihood ratio test.

**Table 3 nutrients-16-00908-t003:** Chi-square test of specific diet and cognitive impairment.

Variables		Cognitive Impairment	χ^2^	*p*
Vegetables	Yes	3516 (27.46)	367.374	<0.001
	No	778 (51.32)		
Fruits	Yes	1632 (24.99)	142.824	<0.001
	No	2662 (34.18)		
Red Meat	Yes	3045 (27.46)	122.027	<0.001
	No	1249 (37.70)		
Fish	Yes	1671 (24.70)	171.180	<0.001
	No	2623 (34.73)		
Eggs	Yes	3003 (29.10)	12.987	<0.001
	No	1291 (32.29)		
Beans–nuts	Yes	2023 (27.47)	45.962	<0.001
	No	2271 (32.66)		
Milk	Yes	1586 (28.37)	11.436	0.001
	No	2708 (31.03)		

**Table 4 nutrients-16-00908-t004:** Chi-square test of dietary diversity scores, patterns, and cognitive impairment.

Variables		Cognitive Impairment	χ^2^	*p*
Dietary Diversity Scores	≤2	1086 (39.63)	260.706 ^a^ 256.236 ^b^	<0.001
	3	748 (33.51)		
	4	851 (31.17)		
	5	757 (26.74)		
	6	546 (24.39)		
	7	306 (19.79)		
Dietary Patterns	DP1	1132 (33.47)	219.698 ^a^ 157.422 ^b^	<0.001
	DP2	1100 (37.68)		
	DP3	1003 (29.91)		
	DP4	1059 (22.71)		

Notes: ^a^ = Pearson’s Chi-square value, ^b^ = linear Chi-square.

**Table 5 nutrients-16-00908-t005:** Binary logistic regression analysis testing the association between the specific diet and cognitive impairment.

Variables	Crude Model		Adjusted Model	
	OR	95% CI	OR	95% CI
Vegetables	0.36	(0.32, 0.40) ***	0.79	(0.72, 0.87) ***
Fruits	0.64	(0.60, 0.69) ***	0.51	(0.45, 0.58) ***
Red Meat	0.63	(0.58, 0.69) ***	0.70	(0.64, 0.78) ***
Fish	0.62	(0.57, 0.66) ***	0.74	(0.68, 0.81) ***
Eggs	0.86	(0.80, 0.93) ***	0.99	(0.90, 1.09)
Beans–nuts	0.78	(0.73, 0.84) ***	0.85	(0.78, 0.93)
Milk	0.88	(0.82, 0.95) ***	0.91	(0.83, 1.00) *

Notes: * *p* < 0.05, *** *p* < 0.001.

**Table 6 nutrients-16-00908-t006:** Binary logistic regression analysis testing the association between the dietary diversity scores, patterns, and cognitive impairment.

Variables	Crude Model		Adjusted Model	
	OR	95% CI	OR	95% CI
Dietary Diversity Scores (ref = ≤ 2)				
3	0.77	(0.68, 0.86) ***	0.79	(0.69, 0.91) **
4	0.69	(0.62, 0.77) ***	0.75	(0.65, 0.86) ***
5	0.56	(0.50, 0.62) ***	0.66	(0.57, 0.75) ***
6	0.49	(0.43, 0.56) ***	0.59	(0.51, 0.69) ***
7	0.38	(0.33, 0.44) ***	0.54	(0.45, 0.65) ***
Dietary Patterns (ref = DP1)				
DP2	1.20	(1.08, 1.33) ***	1.24	(1.09, 1.40) ***
DP3	0.85	(0.77, 0.94) ***	0.91	(0.81, 1.03)
DP4	0.58	(0.53, 0.65) ***	0.79	(0.69, 0.89) ***

Notes: ** *p* < 0.01, *** *p* < 0.001.

**Table 7 nutrients-16-00908-t007:** Pearson correlation analysis of the dietary diversity scores, dietary patterns, psychological balance, and cognitive impairment.

Variable	DDS	DP	Psychological Balance	Cognitive Impairment
DDS	1			
DP	0.862 **	1		
Psychological Balance	0.177 **	0.150 **	1	
Cognitive Impairment	−0.142 **	−0.105 **	−0.233 **	1

Notes: ** *p* < 0.01.

**Table 8 nutrients-16-00908-t008:** Significance test for the mediating effects of dietary diversity scores, psychological balance, and cognitive impairment.

	Effect	BootSE	BootLLCI	BootULCI	Percentage of Total Effect
Total effect	0.302	0.030	0.244	0.361	100%
Direct effect	0.220	0.029	0.163	0.277	73.79%
Dietary diversity scores → psychological balance → cognitive impairment	0.082	0.008	0.066	0.099	27.24%

**Table 9 nutrients-16-00908-t009:** Significance test for the mediating effects of dietary patterns, psychological balance, and cognitive impairment.

	Effect	BootSE	BootLLCI	BootULCI	Percentage of Total Effect
Total effect	0.227	0.044	0.141	0.314	100%
Direct effect	0.134	0.043	0.050	0.218	59.04%
Dietary patterns → psychological balance → cognitive impairment	0.093	0.012	0.070	0.116	41.00%

## Data Availability

The raw data supporting the conclusions of this article can be found at https://opendata.pku.edu.cn/dataverse/CHADS (accessed on 29 January 2024).

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
