# Peer review of "The Mediating Role of Psychological Balance on the Effects of Dietary Behavior on Cognitive Impairment in Chinese Elderly"

_nutrients, 2024, doi:10.3390/nu16060908_

Round 1

Reviewer 1 Report

Comments and Suggestions for Authors

The Manuscript (ID: nutrients-2880981) entitled “The Mediating Role of Psychological Balance on the Effects of Dietary Behavior on Cognitive Impairment in Chinese Elderly: Evidence from the Chinese Longitudinal Healthy Longevity Survey” evaluates an interesting topic on the association between dietary diversity score (DDS), dietary pattern (DP), and cognitive impairment in Chinese elderly. The aim of this study was to provide a basis for developing preventive measures for cognitive impairment and dementia in the elderly subjects.

The main limitation in this Manuscript is the use of the self-reported dietary intake frequency that usually may be associate to bias. In general, the Manuscript is well written and clear to understand, consequently it requires some major revisions.

Specific comments:

The title of the Manuscript should be shortened and focused on the main results.

Line 16, line 21, and line 27 in the Abstract It should be "dietary pattern, and cognitive...".

Lines 36-38. This sentence is probably not correct and must be revised since cognitive impairment is usually considered as a decrease level of cognitive abilities in older adults. Consequently, cognitive impairment is not a higher level of cognition compared to normal older adults.

Usually, the cognitive impairment showed a classification in cognitive normal aging, mild cognitive impairment, and Alzheimer's disease as reported a previous study (Subramanyam Rallabandi et al., 2020. https://doi.org/10.1016/j.imu.2020.100305). It is necessary to better classify the level of cognitive dysfunction evaluated.

In the Materials and Methods section (Lines 110-112), at first Authors should indicate the mean age and standard deviation of their sample and then they could indicate the classification in different age ranges.

Line 160 and Line 163 Authors should indicate for Process 3.5 and Mplus software the complete information. Authors should include the complete description of the software used for their statistical analysis.

Comments on the Quality of English Language

 Minor editing of English language is required

Author Response

Reviewer 1

COMMENTS

The Manuscript (ID: nutrients-2880981) entitled “The Mediating Role of Psychological Balance on the Effects of Dietary Behavior on Cognitive Impairment in Chinese Elderly: Evidence from the Chinese Longitudinal Healthy Longevity Survey” evaluates an interesting topic on the association between dietary diversity score (DDS), dietary pattern (DP), and cognitive impairment in Chinese elderly. The aim of this study was to provide a basis for developing preventive measures for cognitive impairment and dementia in the elderly subjects.

The main limitation in this Manuscript is the use of the self-reported dietary intake frequency that usually may be associate to bias. In general, the Manuscript is well written and clear to understand, consequently it requires some major revisions.

Response: We are very grateful to the reviewer for the hard work and positive comments, Responses to specific comments can be found below. 

Specific comments:

Point 1: The title of the Manuscript should be shortened and focused on the main results.

Response 1: We are very grateful for the judicious and professional comments of the reviewer. We have revised the title of the manuscript accordingly.

Point 2: Line 16, line 21, and line 27 in the Abstract It should be "dietary pattern, and cognitive...".

Response 2: We greatly appreciate the important comment made by the reviewer. We explored the relationship between dietary diversity score and dietary patterns on cognitive impairment, so we used the full name in the first occurrence and abbreviations later.

Point 3: Lines 36-38. This sentence is probably not correct and must be revised since cognitive impairment is usually considered as a decrease level of cognitive abilities in older adults. Consequently, cognitive impairment is not a higher level of cognition compared to normal older adults.

Response 3: We thank the reviewer for pointing this error out and we have revised the statement. We are sorry to admit that there was a mistake in the language expression, and we have rechecked and corrected it. Please review the revised manuscript (Lines 34-36: Cognitive impairment is defined as a decrease level of cognition abilities in older adults, which affects patients' ability to remember, learn, concentrate and make daily decisions).

Point 4: Usually, the cognitive impairment showed a classification in cognitive normal aging, mild cognitive impairment, and Alzheimer's disease as reported a previous study (Subramanyam Rallabandi et al., 2020. https://doi.org/10.1016/j.imu.2020.100305). It is necessary to better classify the level of cognitive dysfunction evaluated.

Response 4:  Because CLHLS used the Mini-Mental State Examination (MMSE). Cognitive impairment in this study was defined as whether the subjects were being neurologically intact. Cognitive impairment in our study may include mild cognitive impairment and different types of dementia. When using CLHLS data for the purpose of analysis, Ren, Z et al. made a similar approach (Ren, Z., et al. https://doi.org/10.1016/j.jad.2021.08.093).

We are very grateful for the professional advice from the reviewer, and we have revised the discussion section with added limitations (Lines 423-425: “Limited by the use of diagnostic tool, our study did not classify cognitive impairment in detail (e.g., mild cognitive impairment, or different types of dementia). ”).

Point 5: In the Materials and Methods section (Lines 110-112), at first Authors should indicate the mean age and standard deviation of their sample and then they could indicate the classification in different age ranges.

Response 5: We thank the reviewer for this important comment. We have revised the article with further details (Line184-185).

Point 6: Line 160 and Line 163 Authors should indicate for Process 3.5 and Mplus software the complete information. Authors should include the complete description of the software used for their statistical analysis.

Response 6: Thank you for the comment. We have added the description of the software used for their statistical analysis (Line 160-162: Dietary patterns were performed in Mplus version 8.3 (Muthen & Muthen, Los Angeles, CA, USA), and correlation and mediation analyses were conducted on SPSS version 25.0 (SPSS Inc., Chicago, IL, USA) at a significance level of 0.05).

Reviewer 2 Report

Comments and Suggestions for Authors

Dear Authors

The paper is well-structured and informative. However, I have a few suggestions to enhance clarity and readability:

Abstract section:

1.     Specify the time frame or period covered by the study to provide context.

2.     Clarify the rationale for using latent class analysis (LCA) in identifying patterns in food varieties

3.     Line 30-31 : Consider rephrasing the last sentence  since you did not specify the time frame. For example “Our findings underscore the importance of promoting a diverse diet from an early age, which may contribute to a lower risk of cognitive impairment in older adult. Furthermore, we do not know what is meant by "early initiation” in your case.

Introduction session:

1.     Line 40 :  “…3% to 53.8%.” I Is this a world or Chinese statistic?

2.     Line 51-52: “Malnutrition and psychological imbalance are two health issues that are becoming more prevalent in older persons. “. It would be more appropriate to stress that these problems are gaining more attention in recent years and not that “becoming more prevalent”.

3.     Line 59: when mentioning the Mediterranean-DASH and MIND diets, briefly explain what these diets consist of for readers who may not be familiar with them and define abbreviations.

4.     Line 89-90: “.. existing research has not yet provided sufficient evidence for the relationship between diet and cognitive impairment.” there are numerous studies and evidence that investigate diet and cognitive impairment. It would be better to express such as “This study investigates the intricate relationship between diet and cognitive impairment”

Materials and Methods section:

1.     Who sponsored the CLHLS in 2018?

2.     Line 110: “age (65-74, 75-84, 85-94 and ≥95) “ . Specify the unit of measurement for age (e.g., years) to ensure clarity.

3.     When explaining the dietary diversity score (DDS) calculation, it is not clear. 

4.     Please, provide a brief rationale for choosing these specific questions from the questionnaire to assess psychological balance to enhance the transparency of your methodology.

5.     Was the Mini-Mental State Examination adapted for the Chinese population? 

Results section:

1.     Table 1. For authors, is it possible to create borders in the table to separate sections (e.g. gender, age etc)? 

2.     Table 1. The last row of table 1 could be misleading. In fact, there is a tendency to sum the column data, leading to an error. Reassess the presence of this data in this row.

3.     Line 177-180.  Sentences are not clear.

Discussion section: 

1.     Provide a concise summary of the main findings related to the association between specific diets, dietary diversity, and cognitive impairment before delving into comparisons with previous studies

2.     The authors should consider how long the subjects have been following this dietary pattern and also discuss this aspect.

3.      Clarify the rationale behind using PB as a mediator and provide a concise explanation of how it contributes to understanding the relationship between dietary factors and cognitive impairment.

Author Response

Reviewer 2

COMMENTS

The paper is well-structured and informative. However, I have a few suggestions to enhance clarity and readability:

Response: Thank you again for your professionalism and conscientiousness.

Abstract section

Point 1: Specify the time frame or period covered by the study to provide context.

Response 1: We are very grateful for the judicious and professional advice of the reviewer. The information that the data used for this study came from CLHLS 2018.

Point 2: Clarify the rationale for using latent class analysis (LCA) in identifying patterns in food varieties

Response 2: Thank you for the comment and LCA has been used in previous studies (Zhang, L. et al. https://doi.org/10.3390/nu15132955). The LCA is a methodological approach that explains population heterogeneity in the data by identifying underlying subgroups of individuals, thus allowing the examination of different DPs while dealing with the diverse nature of the population (Hu, J. et al., 2021.https://doi.org/10.1016/j.ajp.2020.102518). 

Point 3: Line 30-31 : Consider rephrasing the last sentence  since you did not specify the time frame. For example “Our findings underscore the importance of promoting a diverse diet from an early age, which may contribute to a lower risk of cognitive impairment in older adult. Furthermore, we do not know what is meant by "early initiation” in your case.

Response 3: We thank the reviewer for this judicious and professional advice. We have revised the statement. The meant by "early initiation" is a misrepresentation, what we are trying to convey is the need to start dietary diversity as early as possible.

Introduction session

Point 1: Line 40 :  “…3% to 53.8%.” Is this a world or Chinese statistic?

Response 1: We thank the reviewer for this important comment. The prevalence rates of cognitive impairment range from 3% to 53.8% is a world statistic. We have revised the statement in Line 36-37 (Panza, F., et al. doi:10.1176/appi.ajgp.13.8.633.).

Point 2: Line 51-52: “Malnutrition and psychological imbalance are two health issues that are becoming more prevalent in older persons. “. It would be more appropriate to stress that these problems are gaining more attention in recent years and not that “becoming more prevalent”.

Response 2: We thank the reviewer for pointing this error out and we have revised the statement. Please review the revised manuscript (Lines49-50: Malnutrition and psychological imbalance are two health issues affecting older persons that have been gaining more attention in recent years).

Point 3: Line 59: when mentioning the Mediterranean-DASH and MIND diets, briefly explain what these diets consist of for readers who may not be familiar with them and define abbreviations.

Response 3: We thank the reviewer for this important comment. Two assessments of dietary patterns and cognition in non-MCI populations found that the Mediterranean, Dietary Approaches to Stop Hypertension (DASH) and Mediterranean-DASH Intervention for Neurodegenerative Delay (MIND) diets, which are all plant-based diets with moderate to high intakes of fish, are all associated with improved cognitive outcomes. We have revised this part of the content, hoping to be easy for reader to understand (Line 57-62).

Point 4: Line 89-90: “.. existing research has not yet provided sufficient evidence for the relationship between diet and cognitive impairment.” there are numerous studies and evidence that investigate diet and cognitive impairment. It would be better to express such as “This study investigates the intricate relationship between diet and cognitive impairment”

Response 4: We thank the reviewer for this important comment. We have revised the statement (Line 90-91).

Materials and Methods section:

Point 1: Who sponsored the CLHLS in 2018?

Response 1: Thank you for the comment. The study used cross-sectional data from the Chinese Longitudinal Healthy Longevity Survey (CLHLS), which was co-developed by the National Institutes of Development of Peking University's Research Center for Healthy Ageing and Development and the Chinese Center for Disease Control and Prevention in 2018. We have revised the statement (Line 95-98).

Point 2: Line 110: “age (65-74, 75-84, 85-94 and ≥95) “ . Specify the unit of measurement for age (e.g., years) to ensure clarity.

Response 2: Thank you for the comment. We have added the unit of measurement for age (years) (Line 112).

Point 3: When explaining the dietary diversity score (DDS) calculation, it is not clear. 

Response 3: We appreciate comments from the reviewers. We have revised the expression of DDS calculation (Line 129-134: “The DDS calculated the score of adequate intake of seven food groups; values varied from 0 to 7. The highest score is 7, meaning that all seven food groups have been sufficiently intake while the lowest score is 0 indicating that none of the seven groups have been adequately consumed. A higher DDS means more abundant of food groups in adequate amounts intake and more diversity.”).

Point 4: Please, provide a brief rationale for choosing these specific questions from the questionnaire to assess psychological balance to enhance the transparency of your methodology.

Response 4: We thank the reviewer for this important comment. The reason for choosing "Assessment of the current situation and emotional characterization of the personality" to measure PB is that the portion captures older adults' subjective opinions on their general quality of life and standard of living across time. These opinions are relatively stable, and they can be a measure of psychological balance (line 137-140).

Point 5: Was the Mini-Mental State Examination adapted for the Chinese population? 

Response 5: We thank the reviewer for this important comment. The Mini-Mental State Examination adapted for the Chinese population. we have revised the statement (Line 151-152).

Results section:

Point 1: Table 1. For authors, is it possible to create borders in the table to separate sections (e.g. gender, age etc)? 

Response 1: Thank you for the advice. We have created borders in the table to separate sections. Yet We will follow the requirements if there is a need for changes in the editorial office.

Point 2: Table 1. The last row of table 1 could be misleading. In fact, there is a tendency to sum the column data, leading to an error. Reassess the presence of this data in this row.

Response 2: We appreciate comments from the reviewers. In order to avoid misunderstandings, we have removed the last row in Table 1.

Point 3: Line 177-180.  Sentences are not clear.

Response 3: We greatly appreciate the important comment made by the reviewer, and have revised the manuscript (Line 188-193: “Among all the participants, 89.41% of them consumed vegetables daily/ almost daily and frequently, 76.86% of them consumed red meat almost daily/ weekly, 72.08% of them consumed eggs almost daily/ weekly, and 51.44% of them consumed beans-nuts almost daily/ weekly. In contrast, those who intake milk almost daily/ weekly presented low percentages, with 39.04%”).

Discussion section: 

Point 1: Provide a concise summary of the main findings related to the association between specific diets, dietary diversity, and cognitive impairment before delving into comparisons with previous studies

Response 1: We appreciate comments from the reviewers. We have summed up the key conclusions of this study in our discussion of the last sentence in the first paragraph.

Point 2: The authors should consider how long the subjects have been following this dietary pattern and also discuss this aspect.

Response 2: We thank the reviewer for this important and constructive comment. While the time frame was not made explicit when questioning diets, the CLHLS survey restricted the time frame to the previous year when querying responses. We have added some content about the limitation in the Discussion section (Line 429-430: “Moreover, we failed to account for the periods of time that participants adopted specific dietary patterns”).

Point 3: Clarify the rationale behind using PB as a mediator and provide a concise explanation of how it contributes to understanding the relationship between dietary factors and cognitive impairment.

Response 3: We are very grateful for the professional advice from the reviewer. PB may be related to protective factors concerning diet and cognition. There is a relationship between diet and PB, and studies have indicated that that PB is a reliable indicator of cognitive impairment. By comprehending the crucial role of PB, it becomes possible to implement nutritional interventions from a psychological perspective (different from the perspective of previous studies, which mainly explored the factors linked to obesity and metabolism as moderators of diet to cognition) and assist with managing cognitive impairment. The rationale for employing PB as a mediator was covered in the background section. Based on the findings, the discussion section also addressed the part that PB plays in the relationship between nutrition and cognition.

Reviewer 3 Report

Comments and Suggestions for Authors

The study primarily investigates the association between dietary diversity, dietary patterns, and cognitive impairment among the Chinese elderly. It also examines the mediating role of psychological balance in this relationship. The topic is relevant, particularly in the context of aging populations and the increasing prevalence of cognitive impairment. While the relationship between diet and cognitive function is not novel, the focus on psychological balance as a mediator adds a fresh perspective to the field.

Here are some suggestions to improve the manuscript:

- Address the limitations of using cross-sectional data and the implications for causality. Discuss any potential biases introduced by using surrogate respondents.

- Expand on the specific food groups that characterize each dietary pattern. While the manuscript mentions high and low probabilities of consuming certain food groups, a more detailed quantification or description would provide better clarity.

- Expand on the role of psychological balance as a mediator. Discuss why it's a significant mediator, its implications for dietary interventions, and how it compares with other potential mediators in the literature.

- Discuss how your findings can support health initiatives or dietary guidelines, particularly in the context of aging populations in China.

Author Response

Reviewer 3

COMMENTS

The study primarily investigates the association between dietary diversity, dietary patterns, and cognitive impairment among the Chinese elderly. It also examines the mediating role of psychological balance in this relationship. The topic is relevant, particularly in the context of aging populations and the increasing prevalence of cognitive impairment. While the relationship between diet and cognitive function is not novel, the focus on psychological balance as a mediator adds a fresh perspective to the field.

Here are some suggestions to improve the manuscript:

Point 1: Address the limitations of using cross-sectional data and the implications for causality. Discuss any potential biases introduced by using surrogate respondents.

Response 1: We thank the reviewer for this important and constructive comment. We have discussed the limitations of this study in the Discussion section and revised it based on your suggestions (Line 425-429: “This was a cross-sectional study, so reverse causality and bidirectional associations cannot be ruled out. In addition, food frequency questionnaires that rely on self-reported frequency of dietary consumption are vulnerable to recall bias. Bias can also be generated by using surrogate respondents in our study). Please review the revised manuscript.

Point 2: Expand on the specific food groups that characterize each dietary pattern. While the manuscript mentions high and low probabilities of consuming certain food groups, a more detailed quantification or description would provide better clarity.

Response 2: We thank the reviewer for this important comment. We have added Table S1 ( Cluster analysis of seven food groups in the subjects) in the Supplementary Material.

Point 3: Expand on the role of psychological balance as a mediator. Discuss why it's a significant mediator, its implications for dietary interventions, and how it compares with other potential mediators in the literature.

Response 3: A similar comment was raised by reviewer 2. We greatly appreciate the important comment made by the reviewer. PB may be related to protective factors concerning diet and cognition. There is a relationship between diet and PB. Studies have indicated that that PB is a reliable indicator of cognitive impairment. By comprehending the crucial role of PB, it becomes possible to implement nutritional interventions from a psychological perspective (different from the perspective of previous studies, which mainly explored the factors linked to obesity and metabolism as moderators of diet to cognition) and assist with managing cognitive impairment. The rationale for employing PB as a mediator was covered in the background section. Based on the findings, the discussion section also addressed the part that PB plays in the relationship between nutrition and cognition.

Point 4: Discuss how your findings can support health initiatives or dietary guidelines, particularly in the context of aging populations in China.

Response 4: We are very grateful for the professional advice from the reviewer. We have expanded the section on conclusions. Please review the revised manuscript (Line 444-448: Our findings confirm the requirement made by the Chinese Dietary Guidelines for the Elderly to consume a wide variety of foods, including moderate amounts of fruits and vegetables and plenty of fish, poultry, meat, eggs, and beans. This study offers good evidence that encouraging healthy aging can be achieved with a scientific diet).

Thank you again for your professionalism and conscientiousness.

Round 2

Reviewer 1 Report

Comments and Suggestions for Authors

Authors revised the Manuscript in line to Reviewer's suggestion and the quality is improved.

Comments on the Quality of English Language

Minor editing of English language is required

Author Response

We thank the reviewer for the valuable review comments throughout the entire process.

Reviewer 3 Report

Comments and Suggestions for Authors

I would like to thank the Authorts for all improvements they have done. I have no further comments.

Author Response

(The authors gave the same response as above.)
